# Inhibition of Polyamine Catabolism Reduces Cellular Senescence

**DOI:** 10.3390/ijms241713397

**Published:** 2023-08-29

**Authors:** Takeshi Uemura, Miki Matsunaga, Yuka Yokota, Koichi Takao, Takemitsu Furuchi

**Affiliations:** Faculty of Pharmacy and Pharmaceutical Sciences, Josai University, 1-1 Keyakidai, Sakado 350-0295, Saitama, Japanktakao@josai.ac.jp (K.T.); tfuruchi@josai.ac.jp (T.F.)

**Keywords:** aging, senescence, spermine oxidase, polyamine metabolism, acrolein

## Abstract

The aging of the global population has necessitated the identification of effective anti-aging technologies based on scientific evidence. Polyamines (putrescine, spermidine, and spermine) are essential for cell growth and function. Age-related reductions in polyamine levels have been shown to be associated with reduced cognitive and physical functions. We have previously found that the expression of spermine oxidase (SMOX) increases with age; however, the relationship between SMOX expression and cellular senescence remains unclear. Therefore, we investigated the relationship between increased SMOX expression and cellular senescence using human-liver-derived HepG2 cells. Intracellular spermine levels decreased and spermidine levels increased with the serial passaging of cells (aged cells), and aged cells showed increased expression of SMOX. The levels of acrolein-conjugated protein, which is produced during spermine degradation, also increases. Senescence-associated β-gal activity was increased in aged cells, and the increase was suppressed by MDL72527, an inhibitor of acetylpolyamine oxidase (AcPAO) and SMOX, both of which are enzymes that catalyze polyamine degradation. DNA damage accumulated in aged cells and MDL72527 reduced DNA damage. These results suggest that the SMOX-mediated degradation of spermine plays an important role in cellular senescence. Our results demonstrate that cellular senescence can be controlled by inhibiting spermine degradation using a polyamine-catabolizing enzyme inhibitor.

## 1. Introduction

Aging is an irreversible physical change that progresses through complex biological processes [1,2]. Aging is often associated with a decline in physical and cognitive function, resulting in a marked decline in quality of life [3,4]. In addition, the incidence of age-related diseases, such as stroke, dementia, and cardiovascular disease, increases with age [5,6,7,8]. As the global population ages, there is a need for effective scientifically proven anti-aging technologies to prolong the healthy and active lives of people as they age [9].

Age-related decline in physical function has been shown to be caused by the accumulation of senescent cells [10]. Cellular senescence is induced by various stimuli, including oxidative stress, and is characterized by irreversible cell-cycle arrest, macromolecular damage, and impaired cellular metabolism [11]. Cellular senescence is also a potential target for the prevention of physical and cognitive decline in the elderly [12].

Polyamines are bioactive amines that are essential for cell growth and function [13]. Putrescine, spermidine, and spermine are the main polyamines found in mammals [14]. Polyamines are positively charged under physiological conditions and promote cell proliferation by interacting with RNA to stimulate the translation of certain proteins [15,16]. Polyamines have been reported to inhibit age-related DNA methylation changes [17]. It has also been reported that spermidine is essential for eukaryotic translation initiation factor 5A (eIF5A) activity and plays an important role in maintaining mitochondrial function during aging [18]. Because polyamine plays important roles in cellular functions and its levels decrease with age [19], it has been suggested that decline in polyamine levels is related to aging-associated physical changes [20]. Studies using mice, rats, and *Drosophila* have reported that the administration of spermidine can restore cognitive function and extend lifespan [21,22]. In humans, spermidine supplementation has been shown to restore cognitive and immune functions [23,24]. Intracellular polyamine levels are regulated by their biosynthesis, catabolism, and transport [25]. The rate-limiting step in polyamine biosynthesis is putrescine synthesis by ornithine decarboxylase (ODC) [26]. Putrescine receives an aminopropyl group from decarboxylated *S*-adenosylmethionine formed by *S*-adenosylmethionine decarboxylase (AMD1) and is converted into spermidine, which is further converted into spermine [27]. Polyamines are degraded via two pathways: acetylation by spermidine/spermine acetyltransferase, SAT1, followed by degradation by acetylpolyamine oxidase (AcPAO) and direct conversion of spermine to spermidine by spermine oxidase (SMOX) [28,29]. Polyamine synthesis is enhanced in cancer cells [30]. However, the mechanism of age-related polyamine depletion is unclear.

SMOX expression has been previously shown to be upregulated with age in human liver tissue [31]. The degradation of spermine by SMOX is accompanied by the formation of acrolein as a byproduct [32,33]. Acrolein is a highly reactive unsaturated aldehyde that adds to nucleic acids and cytoskeletal proteins, disrupting their function and causing cytotoxicity [34,35,36]. However, the effect of SMOX expression in polyamine levels in aging and the relationship between the age-related increments in polyamine degradation and aging remains unclear.

In the present study, we investigated the relationship between polyamine metabolism and senescence in HepG2 cells.

## 2. Results

### 2.1. Effect of Long-Term Culture on Cellular Polyamine Levels and Polyamine Metabolism Enzyme Levels

To examine the effect of aging on cellular polyamine metabolism, HepG2 cells were cultured for three months with passaging performed every three days (aged cells), and the polyamine content in these cells was compared with that in cells with a low passage number (young cells). Figure 1 shows the polyamine levels in aged and young cells. Aged cells showed higher spermidine levels and lower spermine levels than young cells (Figure 1). However, when the cells were cultured with MDL72527, an inhibitor of AcPAO and SMOX that catalyze polyamine degradation [37], spermidine levels decreased and spermine levels increased in both young and aged cells (Figure 1). Thus, spermine degradation was activated in aged cells. Putrescine was not detected in any of the cells.

Next, we measured the levels of polyamine-metabolizing enzymes. A polyamine-metabolizing pathway is summarized in Figure 2A. As shown in Figure 2B,C, SMOX was induced in aged cells, and MDL72527 treatment did not affect SMOX levels. ODC levels were increased in aged cells, whereas AMD1 levels were unchanged. Spermidine and spermine synthase levels were unaltered. SAT1 and AcPAO levels decreased in aged cells. The levels of protein-conjugated acrolein (PC-Acro) increased in aged cells, and MDL72527 treatment decreased PC-Acro levels. Glyceraldehyde 3-phosphate dehydrogenase (GAPDH) expression was slightly decreased in aged cells, and MDL72527 treatment rescued this decrease. This decrease in GAPDH levels is consistent with a previous report by another group [38]. These results indicate that cell aging increased SMOX levels and spermine degradation.

### 2.2. Effect of Polyamine-Catabolizing Enzyme Inhibitor on Cellular Senescence

To clarify whether the age-associated increase in spermine degradation contributes to cellular senescence, senescence associated β-gal activity was measured. As shown in Figure 3, senescence associated β-gal activity was significantly higher in aged cells than in young cells. MDL72527 treatment suppressed the increase in senescence associated β-gal activity in aged cells. The level of p21, which has been reported to increase in senescent cells [39], was higher in aged cells than in young cells, and MDL72527 treatment reversed this increase (Figure 2B). The level of p16, which is also upregulated in senescent cells [40], was unchanged in aged cells (Figure 2B). The induction of p21 occurs prior to the induction of p16 [41]. These results indicate that spermine degradation plays an important role in the early stage of cellular senescence.

### 2.3. Effect of Polyamine Metabolism Inhibitor on DNA Damage Induced by Cellular Aging

DNA damage was examined using a DNA Damage Detection Kit. The kit detects phosphorylated γH2AX, which accumulates in damaged DNA [42]. As shown in Figure 4, DNA damage was not detected in young treated or non-treated cells. However, DNA damage accumulated in untreated aged cells. MDL72527-treated cells showed reduced signals from the damaged DNA.

## 3. Discussion

In this study, we used a culture of HepG2 cells as a model to investigate the relationship between changes in polyamine metabolism and cellular senescence during aging. To analyze the effects of cellular senescence, HepG2 cells were cultured with passaging for 3 months. Polyamine content was significantly altered in aged cells (Figure 1). The spermine levels decreased and spermidine levels increased in aged cells, suggesting that conversion of spermine to spermidine is activated as cells age. Western blotting showed that SMOX levels increased in aged cells (Figure 2B,C). In contrast, SAT1 and AcPAO levels decreased in aged cells. No changes were observed in the expression levels of AMD1, spermidine synthase, SPDS and spermine synthase, SPMS. These results suggest that the decrease in spermine levels and increase in spermidine levels are due to increased SMOX activity and that the contribution of spermidine synthesis or degradation of spermine by the SAT1/AcPAO pathway may be small. This suggestion was supported by an increase in the protein-conjugated acrolein levels in the aged cells (Figure 2B,C). The degradation of spermine by SMOX produces acrolein as a byproduct, leading to an increase in protein-conjugated acrolein levels [43,44]. Acrolein conjugation inactivates protein function and induces DNA damage [45,46]. As shown in Figure 4, DNA damage increased with age; however, this accumulation was suppressed by MDL72527 treatment. One protein affected by acrolein conjugation is GAPDH [47]. Acrolein conjugation promoted GAPDH degradation. As shown in Figure 2, GAPDH levels were decreased in aged cells, and this decrease was reversed by MDL72527 treatment, indicating that acrolein is produced during aging and that GAPDH was subjected to degradation by acrolein conjugation.

We found that p21 was increased in aged cells and the increase was suppressed by MDL72527 treatment. However, the p16 level was unchanged in aged cells. p21 is elevated in cell-cycle arrest during senescence [48] because it induces and maintains senescence [41]. These findings suggest that SMOX-mediated degradation of spermine plays an important role in the early stage of senescence.

Notably, the amount of spermidine was higher in the aged cells than in the young cells (Figure 1). Several reports have shown that spermidine restores mitochondrial function and the senescence phenotype [17,18,22,24,49,50,51]. Our data show that increased spermidine levels do not suppress senescence. This result indicates that polyamine degradation may contribute more to senescence than age-related reductions in spermidine levels. It is possible that suppression of spermine degradation, in addition to spermidine supplementation, may be more effective in delaying senescence.

In addition to the metabolic pathway, cellular polyamine levels are also regulated by transport [25]. To further understand the effect of altered polyamine levels in cellular ageing, it is important to elucidate the role of polyamine transport in the regulation of cellular polyamine levels with cellular ageing.

The mechanism by which SMOX expression increases and SAT1 and AcPAO expression decreases with age, and its physiological significance, remains unclear. Although this study used a long-term culture model of cultured cells, it is unclear to what extent it reflects cellular senescence in vivo. Further studies are required to elucidate the role of polyamine catabolism in aging.

## 4. Materials and Methods

### 4.1. Reagents

Trichloroacetic acid, high-performance liquid chromatography-grade acetonitrile, and water were purchased from Fujifilm Wako (Saitama, Japan). The polyamine-catabolizing enzyme inhibitor, MDL72527, was synthesized according to a previously reported method [52].

### 4.2. Cell Culture

Human hepatocyte carcinoma-derived HepG2 cells, provided by RIKEN BRC through the National Bio-Resource Project of the MEXT, Japan, were cultured in minimum essential medium supplemented with 50 U/mL streptomycin, 100 U/mL penicillin G (Nacalai tesque), and 10% fetal calf serum at 37 °C in an atmosphere with 5% CO_2_. Aged cells were generated by serially passaging the cells for 3 months. Cells subjected to fewer than three passages after thawing were used as young cells. For MDL72527 treatment, 20 µM of MDL72527 was added to the culture medium during 3 months of culture.

### 4.3. Measurement of Polyamine Content

Polyamine levels in the cells were determined by ion-pair high-performance liquid chromatography using the method described by Vidal-Carou et al. [53] with modifications. Briefly, the cells were washed three times with ice-cold phosphate-buffered saline (PBS) and homogenized in RIPA buffer (Nacalai Tesque, Kyoto, Japan). Polyamines were extracted by adding trichloroacetic acid to a final concentration of 5% and incubation at 90 °C for 30 min. After centrifugation, the supernatants were collected and analyzed using a Jasco HPLC system with an InertSustain C18 column (particle size, 3 µm; internal diameter, 4.6 mm; length, 150 mm; GL Science). The cellular protein content was determined using a BCA protein assay kit (Nacalai Tesque) with bovine serum albumin (Nacalai Tesque) as the standard.

### 4.4. Western Blotting

The cells were washed with PBS and homogenized in RIPA buffer containing a protease inhibitor cocktail (Nacalai Tesque, Japan). Proteins were separated on a 10% polyacrylamide gel and electrophoretically transferred to an immunoblotting polyvinylidene difluoride (PVDF) membrane (Bio-Rad Laboratories, Inc., Hercules, CA, USA). The blots were blocked with the Blocking One reagent (Nacalai Tesque) for 30 min at room temperature. SMOX, ODC, AMD1, SPDS, SPMS, SAT1, AcPAO, protein-conjugated acrolein, p21, p16, GAPDH and actin were detected with a Chemi-Lumi One reagent (Nacalai tesque) using anti-SMOX (Proteintech), anti-ODC (Proteintech), anti-AMD1 (Proteintech), anti-SPDS (Proteintech), anti-SPMS (Proteintech), anti-SAT1 (abcam), anti-AcPAO (abcam), anti-protein-conjugated acrolein (abcam), anti-p21 (Proteintech), anti-p16 (Proteintech), anti-GAPDH (invitorgen) and anti-β-actin (MBL) antibodies. Images were acquired using an ImageQuant LAS 500 (GE Healthcare). The bands were quantified using the ImageJ program [54], normalized to Actin and expressed as relative amount of young, non-treated cells.

### 4.5. Measurement of Senescence-Associated β-gal Activity

Senescence-associated β-gal activity was measured using a Cellular Senescence Plate Assay Kit (Dojindo, Tokyo, Japan) according to the technical manual provided by the manufacturer. Twenty thousand cells were seeded on a black Fluotrac plate (Greiner Bio, Frickenhausen, Germany) and cultured overnight. DNA was stained with Hoechst 33342 and quantified using a Cell Count Normalization Kit (Dojindo, Japan) before the β-gal assay. Fluorescence was measured using SpectraMax M5e (Molecular Devices, Tokyo, Japan). The β-gal activity was normalized to the Hoechst intensity.

### 4.6. Detection of DNA Damage

DNA damage was detected using the DNA Damage Detection Kit-Red (Dojindo, Japan) according to the manufacturer’s protocol. Cells were cultured on coverslips, washed with phosphate-buffered saline (PBS), and fixed in 2% paraformaldehyde for 15 min at 37 °C. The coverslips were immersed in cold acetone for 15 s before staining. After staining, the cells were mounted in the ProLong Diamond Mounting Solution with DAPI (Invitrogen). Fluorescence was visualized using a BZ-X810 fluorescence microscope (Keyence, Osaka, Japan). The intensity of red fluorescence in cells was quantified using the ImageJ program. At least 10 cells each from three independent staining were measured.

### 4.7. Statistical Analysis

Statistical analyses were performed using R software, version 4.3 (R Core Team (2018). R: Language and environment for statistical computing. R Foundation for Statistical Computing, Vienna, Austria. URL https://www.R-project.org/ (accessed on 1/August/2023)), and figures were generated using ggplot2 [55]. Differences between groups were compared using Tukey’s multiple comparison test. A *p*-value < 0.05 was considered statistically significant.

## Figures and Tables

**Figure 1 ijms-24-13397-f001:**
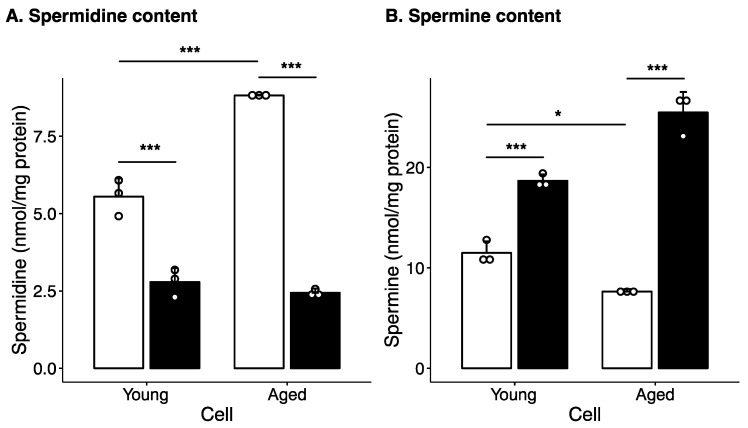
Effect of aging and MDL72527 on polyamine levels. HepG2 cells cultured in the absence (open column) or presence (filled column) of 20 µM MDL72527 for 3 days (young) and 3 months with serial passage (aged). Polyamine levels were measured as described in the Materials and Methods section. Spermidine (**A**) and spermine (**B**) levels were calculated as nmol/mg protein and expressed as mean ± standard deviation (SD) of triplicate determinations. The dots indicate individual data points. * *p* < 0.05, *** *p* < 0.005.

**Figure 2 ijms-24-13397-f002:**
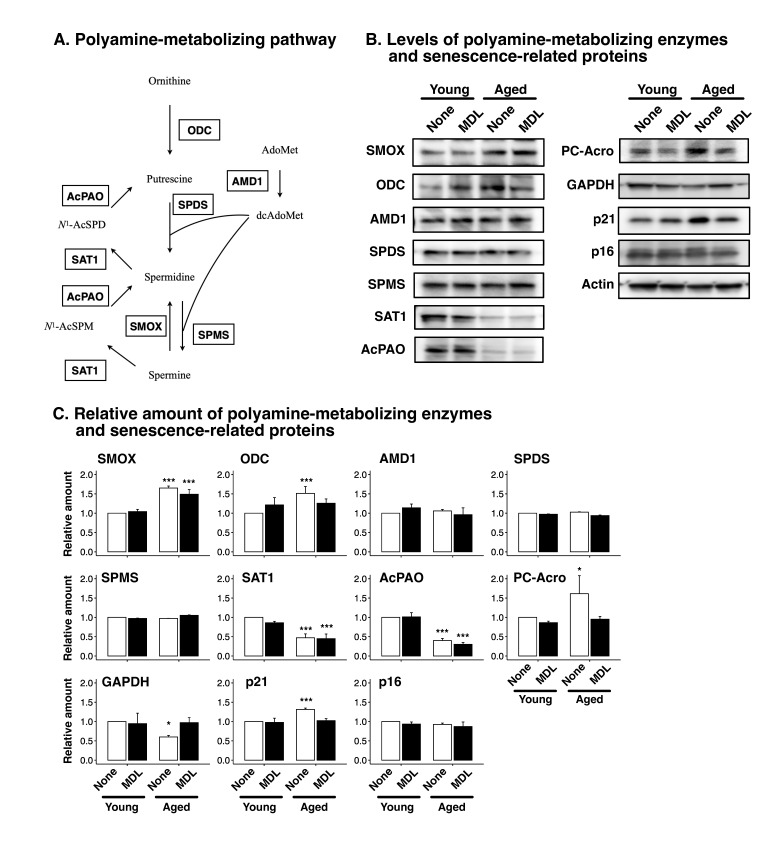
Effect of aging on the protein levels of polyamine-metabolizing enzymes and senescence-related proteins. (**A**) Polyamine-metabolizing pathway. AdoMet; *S*-adenosylmethionine, dcAdoMet; decarboxylated *S*-adenosylmethionine, *N*^1^-AcSPD; *N*^1^-acetylspermidine, *N*^1^-AcSPM; *N*^1^-acetylspermine. (**B**) Cells were cultured in the absence (none) or presence (MDL) of 20 µM MDL72527 for 3 days (young) and serially passaged for 3 months (aged). Western blotting was performed as described in the Materials and Methods. The full-length blots are shown in Appendix A. (**C**) Bands were quantified as described in Materials and Methods, normalized to actin and expressed as relative amount to untreated young cells. Data are presented as the mean + SD of triplicate determinations. * *p* < 0.05, *** *p* < 0.005.

**Figure 3 ijms-24-13397-f003:**
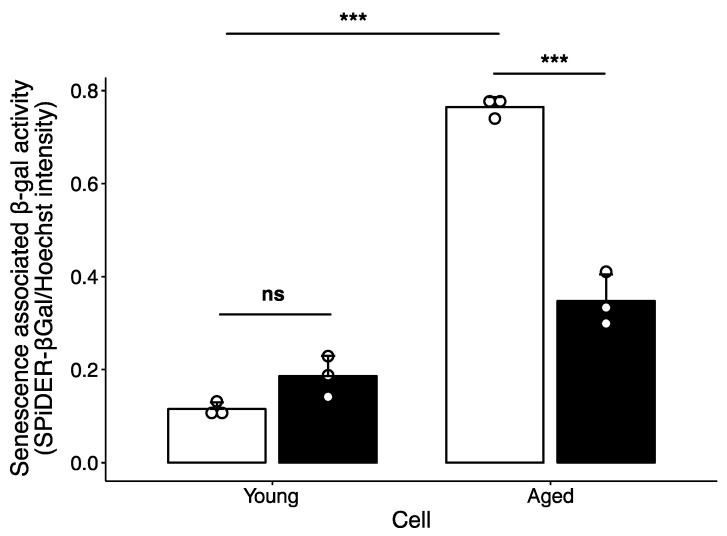
Effect of aging and MDL72527 on senescence-associated β-gal activity. Cells were cultured in the absence (open column) or presence (filled column) of 20 µM MDL72527 for 3 days (young) or 3 months with serial passaging (aged) and seeded on a black 96-well plate. Senescence-associated β-gal activity was measured as described in the Materials and Methods. Data are presented as the mean ± SD of triplicate measurements. Dots indicate individual measurements. ns, not significant; *** *p* < 0.005.

**Figure 4 ijms-24-13397-f004:**
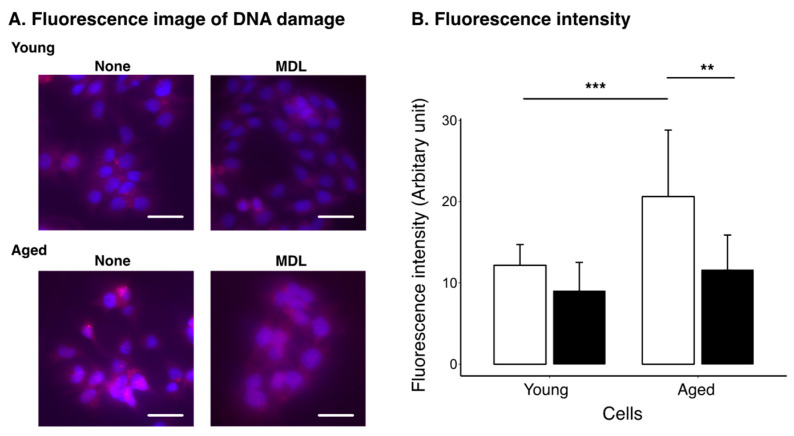
Effect of aging and MDL72527 on DNA damage. (**A**) Cells were cultured in the absence (none) or presence (MDL) of 20 µM MDL72527 for 3 days (young) and serially passaged for 3 months (aged). Cells were seeded on coverslips, cultured for 1 d, and fixed. DNA damage was visualized as described in the Materials and Methods section. Red fluorescence indicates phospho-γH2AX, and blue fluorescence indicates nuclei. Bar = 20 μm. (**B**) Red fluorescence intensity in cells was measured as described in the Materials and Methods section and shown as mean + SD of at least 10 cells from three independent staining. Open columns and filled columns indicate none and MDL, respectively. ** *p* < 0.01, *** *p* < 0.005.

## Data Availability

The datasets generated and/or analyzed in the current study are available from the corresponding author upon reasonable request.

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
