# Peer review of "Inhibition of Polyamine Catabolism Reduces Cellular Senescence"

_ijms, 2023, doi:10.3390/ijms241713397_

Round 1

Reviewer 1 Report

In this paper, the authors addressed the association between the polyamine metabolism and senescence by studying aged cells vs. young cells. They found that there is concomitant increase in spermidine and decrease in spermine levels in aged cells vs. young cells that was abrogated by polyamine metabolizing enzyme inhibitor MDL72527. Further measurements of polyamine metabolizing enzymes by western blot  showed increased SMOX protein and protein-conjugated acrolein levels. Furthermore they show increased B-gal activity and DNA damage in aged cells relative to young cells.

The manuscript is very difficult to understand as the result section does not match with the figures. For example the 1A figure with filled columns (black) showed no change in spermidine levels between aged vs young cells without inhibitor in the legend. But the results section mention increased spermidine levels in aged cells, which is actually for opened columns.

The authors mentioned MDL72527 as inhibitor of polyamine metabolizing enzymes. It is not clear whether it inhibits synthesis or degradation polyamines. Without this information, it is difficult to understand and evaluate the manuscript.

The introduction of the manuscript is not complete and lacks many important details that are required to understand the results. More emphasis should be given on the role of polyamines role on cellular functions and regulatory pathways or conditions that control their cellular levels.

The authors do not discuss the decrease in SAT1 and AcPAO levels in aged cells and  their effect on polyamine levels.

The first sentence in the abstract (line 7-8) and sentence in line 29-30 are exactly same. Authors should rewrite the sentence to avoid repeat sentences.

The manuscript is written in oversimplified manner. More thought should be given to emphasis the results and conclusions rather than just informing there is increase or decrease in the levels of the testing parameter. The authors should take care not to repeat the sentences in the manuscript. 

Author Response

Responses to Reviewer #1

 We would like to thank the reviewer for the effort and valuable comments.

Comment 1-1: The manuscript is very difficult to understand as the result section does not match with the figures. For example the 1A figure with filled columns (black) showed no change in spermidine levels between aged vs young cells without inhibitor in the legend. But the results section mention increased spermidine levels in aged cells, which is actually for opened columns.

Response 1-1: Thank you for pointing them out. We corrected mismatches in the figure legends for figures 1 and 3.

Comment 1-2: The authors mentioned MDL72527 as inhibitor of polyamine metabolizing enzymes. It is not clear whether it inhibits synthesis or degradation polyamines. Without this information, it is difficult to understand and evaluate the manuscript.

Response 1-2: It is clearly stated that MDL72527 is an inhibitor of polyamine degrading enzymes (lines 77-78). The term “polyamine-metabolizing enzyme inhibitor” was changed to “polyamine-catabolizing enzyme inhibitor” (lines 21, 112 and 191).

Comment 1-3: The introduction of the manuscript is not complete and lacks many important details that are required to understand the results. More emphasis should be given on the role of polyamines role on cellular functions and regulatory pathways or conditions that control their cellular levels.

Response 1-3: The introduction was revised to include detailed role of polyamines in cellular functions related to aging and conditions that affect their cellular levels (lines 40-47, 58-59).

Comment 1-4: The authors do not discuss the decrease in SAT1 and AcPAO levels in aged cells and their effect on polyamine levels.

Response 1-4: Thank you for the comment. The decrease in SAT1 and AcPAO in aged cells and their effect of polyamine levels were discussed in lines 159-160.

Comment 1-5: The first sentence in the abstract (line 7-8) and sentence in line 29-30 are exactly same. Authors should rewrite the sentence to avoid repeat sentences.

Response 1-5: The sentence in line 30-32 was rewritten to avoid repeat sentences.

Comments on the Quality of English Language: The manuscript is written in oversimplified manner. More thought should be given to emphasis the results and conclusions rather than just informing there is increase or decrease in the levels of the testing parameter. The authors should take care not to repeat the sentences in the manuscript.

Response : Thank you for the comment. The manuscript was revised to avoid oversimplification. Changes are shown in red letters. We double checked not to repeat the sentences. English was edited by Editage English language editing service and a confirmation certificate was sent to the editorial office.

Author Response

Responses to Reviewer #2

We would like to appreciate the effort and valuable comments of the reviewer.

Comment 2-1: Line 82, can the author spell out AMD1 and give a brief introduction of this enzyme in the introduction part?

Response 2-1: Thank you for the comment. AMD1 was spelled out and the role of AMD1 was explained in Introduction line 53-54.

Comment 2-2: Figure 2, p16 lane was not explained. What’s its function in cell senescence?

Response 2-2: The level of p16 in cells and a role of p16 in cellular senescence was explained and discussed in lines 118-121 and 170-174.

Comment 2-3: I suggest the author to use a scheme to show the polyamine synthesis and degradation pathway and indicate all related enzymes (mentioned in this manuscript).

Response 2-3: Thank you for the suggestion. The scheme for polyamine synthesis and degradation pathway was added in figure 2A.

Round 2

Reviewer 1 Report

Major comments:

Figure 2 and 4: I assume the western blot and florescence experiments were conducted in replicates. Quantity the changes in fold changes for each protein with error bars and p-value. Do the same for figure 4. P-value is compulsory to show the significant changes.

Minor comments:

The authors modified the statement “MDL72527, an inhibitor of enzymes catalyzing polyamine degradation”. The statement needs further clarification as three enzymes SMOX, SAT1 and AcPAO are involved in conversion of spermine to spermidine by two different pathways. The authors should explicitly mention which enzyme does MDL72527 inhibits.

Line 43: What is eIF5a?

Line 114: b-gal activity should be replaced with senescence associated b-gal activity.

The authors didn’t include the effects of polyamine transport in young vs aged cells. They should at least comment on the role of polyamine transport in aging.

 Line 158-159: It appears the modified statement was introduced haphazardly. It disrupts, otherwise the continues statement in the previous version of the manuscript. Need to rewrite the passage.

Author Response

Responses to Reviewer #1

 We would like to thank the reviewer for the valuable comments and advise.

Major comments:

Figure 2 and 4: I assume the western blot and florescence experiments were conducted in replicates. Quantity the changes in fold changes for each protein with error bars and p-value. Do the same for figure 4. P-value is compulsory to show the significant changes.

Response: Thank you for the advice. The quantifications were performed and Figures 2 and 4 were updated with p-values.

Minor comments:

The authors modified the statement “MDL72527, an inhibitor of enzymes catalyzing polyamine degradation”. The statement needs further clarification as three enzymes SMOX, SAT1 and AcPAO are involved in conversion of spermine to spermidine by two different pathways. The authors should explicitly mention which enzyme does MDL72527 inhibits.

Response: the statement was modified to “MDL72527, an inhibitor of acetylpolyamine oxidase (AcPAO) and SMOX, both of which are enzymes that catalyze polyamine degradation”, line 17 and “an inhibitor of AcPAO and SMOX that catalyze polyamine degradation”, line 77.

Line 43: What is eIF5a?

Response: line 43 was changed to “eukaryotic translation initiation factor 5A (eIF5A)”.

Line 114: b-gal activity should be replaced with senescence associated b-gal activity.

Response: β-gal activity was replaced with “senescence associated β-gal activity”.

The authors didn’t include the effects of polyamine transport in young vs aged cells. They should at least comment on the role of polyamine transport in aging.

Response: The role of polyamine transport in the regulation of cellular polyamine levels with aging was mentioned in discussion, lines 190-193.

Comments on the Quality of English Language:

Line 158-159: It appears the modified statement was introduced haphazardly. It disrupts, otherwise the continues statement in the previous version of the manuscript. Need to rewrite the passage.

Response: Thank you for the comment. The statement was rewritten, lines 165-168.